# Phenotypic Expression and Outcomes in Patients with the p.Arg301Gln *GLA* Variant in Anderson–Fabry Disease

**DOI:** 10.3390/ijms25084299

**Published:** 2024-04-12

**Authors:** Rocío Blanco, Yolanda Rico-Ramírez, Álvaro Hermida-Ameijeiras, Israa Mahmoud Sanad Abdullah, Kolja Lau, Jorge Alvarez-Rubio, Elena Fortuny, Amparo Martínez-Monzonís, Albina Nowak, Peter Nordbeck, Carlos Veras-Burgos, Jaume Pons-Llinares, Emiliano Rossi, Fiama Caimi-Martínez, Teresa Bosch-Rovira, Marta Alamar-Cervera, Virginia Ruiz-Pizarro, Laura Torres-Juan, Damian Heine-Suñer, Tomás Ripoll-Vera

**Affiliations:** 1Cardiology Department, Hospital Universitario Son Llatzer, 07198 Palma de Mallorca, Spain; rocio.blanco@hospitaliltaliano.org.ar (R.B.); jalvarezr@hsll.es (J.A.-R.); malamar@hsll.es (M.A.-C.);; 2Cardiology Department, Italian Hospital of Buenos Aires, Buenos Aires C1199ABB, Argentina; emiliano.rossi@hospitalitaliano.org.ar; 3Cardiology Department, Hospital Universitario Son Espases, 07120 Palma de Mallorca, Spain; yolanda.rico@ssib.es (Y.R.-R.); elena.fortuny@ssib.es (E.F.); jaumea.pons@ssib.es (J.P.-L.);; 4Department of Internal Medicine, Clinical University Hospital of Santiago de Compostela, 15706 Santiago de Compostela, Spain; alvaro.hermida.ameijeiras@sergas.es; 5Department of Endocrinology and Clinical Nutrition, University Hospital Zurich, 8091 Zurich, Switzerland; israa.abdullah@mail.ch (I.M.S.A.); albina.nowak@usz.ch (A.N.); 6Division of Internal Medicine, Psychiatric University Hospital Zurich, 8008 Zurich, Switzerland; 7Department of Internal Medicine I, University Hospital Würzburg, 97080 Würzburg, Germany; lau_k@ukw.de (K.L.); nordbeck_p@ukw.de (P.N.); 8The Health Research Institute of the Balearic Islands (IdISBa), 07120 Palma de Mallorca, Spain; laura.torresjuan@ssib.es (L.T.-J.);; 9Cardiology Department, Clinical University Hospital of Santiago de Compostela, 15706 Santiago de Compostela, Spain; ampafm4@gmail.com; 10Centro de Investigación en Red de Enfermedades Cardiovasculares (CIBERCV), 28029 Madrid, Spain; 11Molecular Diagnostics and Clinical Genetics Unit, Hospital Universitario Son Espases, 07120 Palma de Mallorca, Spain; 12Biomedical Research Networking Center for Physiopathology of Obesity and Nutrition (CIBEROBN), Carlos III Health Institute, 28029 Madrid, Spain

**Keywords:** Anderson–Fabry disease, AFD, α-galactosidase A deficiency, renal failure, lysosomal, hypertrophic cardiomyopathy

## Abstract

The p.Arg301Gln variant in the α -galactosidase A gene (*GLA*) has been poorly described in the literature. The few reports show controversial information, with both classical and nonclassical Anderson–Fabry Disease (AFD) presentation patterns. The aim of this study was to analyze the penetrance, clinical phenotype, and biochemical profile of an international cohort of patients carrying the p.Arg301Gln genetic variant in the *GLA* gene. This was an observational, international, and retrospective cohort case series study of patients carrying the p.Arg301Gln variant in the *GLA* gene associated with AFD disease. Forty-nine p.Arg301Gln *GLA* carriers, 41% male, were analyzed. The penetrance was 63% in the entire cohort and 1.5 times higher in men. The mean age of symptoms onset was 41 years; compared to women, men presented symptoms earlier and with a shorter delay to diagnosis. The typical clinical triad—cornea verticillate, neuropathic pain, and angiokeratomas—affected only 20% of the cohort, with no differences between genders. During follow-up, almost 20% of the patients presented some type of nonfatal cardiovascular and renal event (stroke, need for dialysis, heart failure, and arrhythmias requiring intracardiac devices), predominantly affecting men. Residual levels were the most common finding of α-GAL A enzyme activity, only a few women had a normal level; a small proportion of men had undetectable levels. The incidence of combined outcomes including all causes of death was 33%, and the cumulative incidence of all-cause mortality was 9% at the follow-up. Patients carrying the p.Arg301Gln *GLA* variant have a high penetrance, with predominantly cardiorenal involvement and clinical onset of the disease in middle age. Only a small proportion showed the classic clinical presentation of AFD. As in other X-linked diseases, males were more affected by severe cardiovascular and renal events. This genotype–phenotype correlation could be useful from a practical clinical point of view and for future decision making.

## 1. Introduction

Anderson–Fabry Disease (AFD) disease is the result of genetic variants that affect the correct functioning of the enzyme α-galactosidase A (α-GAL A), which causes the abnormal deposition of glycosphingolipids at the lysosomal level and affects multiple tissues and organs. The concentration and enzyme’s functional degree varies not only depending on the type of genetic change, but also on the number of mutated alleles, and due to X-linked genetic transmission, a heterogeneity of presentation is almost invariably observed between males and females.

From the molecular point of view, the *GLA* gene consists of seven exons. Like in other genetic cardiomyopathies, some ‘hot spots’ for point mutations have been identified in this gene, particularly those CpG and missense variants impacting the active site or key residues of the enzyme [1,2,3]. In particular, the p.Arg301Gln variant (NM_000169.2:c.902G>A) generates a missense-type alteration of the genetic code that replaces the arginine at position 301 of exon 6 with a glutamine, generating a partially conservative change of amino acids in the coded protein sequence [3]. A study by Lien et al. determined that this alteration does not affect the enzyme’s catalytic function but does interfere with intracellular transport between the endoplasmic reticulum and the cis-Golgi compartment, resulting in degradation of the defective enzyme [4].

From a clinical point of view, a few reports in the literature have shown that p.Arg301Gln is mainly associated with a midlife onset of clinical manifestations, with less extensive organ involvement, corresponding to nonclassical variants. However, it is believed that it could also have a classical presentation [5,6]. Unlike patients with the typical presentation, who usually have undetectable or very low levels of α-GAL A activity, the nonclassical variants preserve some residual enzyme activity in both men and women [5,7,8]. Although the functional clinical impact of each genetic variant is multifactorial, the development of the clinical phenotype depends not only on the presence of homozygosity or hemizygosity or the phenomenon of gene silencing (lionization) in early embryogenesis in women, but also on the specific type of genetic variant and whether this modification in the conformation of the protein structure partially conserves some degree of enzymatic activity [7,9]. Taking this into account, there is a relationship between the type of genetic variant and the severity and timing of the development of the disease, whether early-onset and classical or late-onset, nonclassical. The balance between the rate of glycosphingolipid production and substrate accumulation, associated with the degree of tolerance of each tissue to such accumulation, will determine the functional impact on each organ and individual patient. Likewise, and as occurs for other cardiomyopathies of genetic origin, classic cardiovascular risk factors could potentiate the effect of these genetic variants on organ damage [10].

Since de novo mutations only occur in 3 to 10% of cases [11], it is considered mainly a familial disease, so cascade testing of relatives is mandatory to be able to analyze the genotype–phenotype correlation in AFD.

The introduction of long-term enzyme replacement therapy (ERT) has led to significant changes in disease progression [3]. Current treatment options are recombinant ERT with intravenous agalsidase-alfa or agalsidase-beta every 2 weeks and oral chaperone therapy with migalastat. The chaperones stabilize and facilitate the trafficking of amenable mutant forms of α-GAL A enzyme from the endoplasmic reticulum to lysosomes and increase its lysosomal activity [12]; these are administered orally every two days and are generally well-tolerated. On the other hand, ERT is administered intravenously, twice a month, and there is a high proportion of infusion associated [13], so they must be premedicated. The decision of which to choose depends not only on the tolerability of the intravenous infusion, but also on the type of genetic change, the consequences on basal enzyme activity, and the degree of improvement after the administration of migalastat.

It is noteworthy that, like a few other genetic variants of this gene, p.Arg301Gln is considered ‘amenable’ to pharmacological treatment with oral chaperones, which broadens the therapeutic strategies for these patients [14]. The amenability is defined as an absolute increase in enzyme activity of ≥3% of wild type enzyme activity and a relative increase in enzyme activity of ≥1.2-fold achieved following chaperone treatment [15].

As in other genetic-based diseases, genotype–phenotype correlation is still poorly understood. The aim of this study was to describe the clinical characteristics, penetrance, degree of expressivity, and biochemical profile of a multinational cohort of patients carrying the p.Arg301Gln genetic variant in the *GLA* gene that is associated with AFD.

## 2. Results

Forty-nine patients carrying the p.Arg301Gln genetic variant in the *GLA* gene were evaluated, 41% of whom were male. Only 4%, two females, were considered obligate carriers given the X-linked inheritance pattern. Seventy percent were detected in relation to family screening, 9% were studied for some suggestive alteration in complementary studies, and only 21% due to the presence of characteristic symptoms of this disease.

The median follow-up was 3 years and 6 months (IQR 2 years and 11 month–4 years and 11 months).

The overall proportion of asymptomatic carriers was 37%, representing 20% of the male and 48% of the female group. Sixty-three percent had some clinical manifestation of the disease, with a penetrance 1.54 times higher in men than in women (80% vs. 52%; *p* = 0.044) and a symptomatic/asymptomatic ratio of 4 and 1.07, respectively (Figure 1a).

The median age at diagnosis was 41 years (IQR 21–56), and 46 years (IQR 36–59) for the symptomatic subgroup. The delay to onset of symptoms was 10 years later in females than in males (46 vs. 37 years). The age and penetrance distribution between the groups is shown in Figure 1b. 

Ten percent of patients were detected before the development of symptoms. Excluding these, the average diagnostic delay from symptom onset was 13 months (IQR 6 months–12 years). Females presented more delay in diagnosis from the onset of symptoms, and a few were diagnosed before the beginning of symptoms.

### 2.1. Organ Involvement and Differences according to Gender

The clinical event incidence and distribution by gender are shown in Figure 2.

The most affected organs in both men and women were the heart and kidneys. About 40% of patients presented proteinuria; less than 10% evolved during follow-up to end-stage renal failure requiring renal replacement therapy.

Thirty percent developed heart failure symptoms with dyspnea at a New York Heart Association functional class (NYHA) ≥ 2.

The most common electrocardiographic findings were T-wave inversion and left ventricular hypertrophy (LVH) signs (Table 1). Bundle branch blocks and LVH were statistically more frequent in men, whereas the presence of AV 1 block (AVB) and short PR interval did not differ between genders.

The occurrence of arrhythmias such as atrial fibrillation, advanced AVB, non-sustained ventricular tachycardia (NSVT), and pacemaker placement affected 20% of the cohort. The most prevalent arrhythmia was atrial fibrillation, with 14% incidence during the follow-up and was more prevalent in males. Conduction disturbances were not frequent, with a 10% incidence of bundle branch blocks, and pacemaker placement was necessary in 6% of the cohort. Both NSVT (incidence of 20%, *p* = 0.013), sustained VT (incidence of 10%, *p* = 0.087) as well as pacemaker implantation (*p* = 0.031) only occurred in men, requiring upgrading of the device to implantable automatic defibrillator at follow-up in two of these patients.

Eight percent presented an acute cerebrovascular event during follow-up.

Multiorgan damage, beyond cardiorenal involvement, was relatively rare. Only 20% of patients presented the classic triad of the disease such as cornea verticillata, neuropathic pain, and angiokeratomas. Acroparesthesias and gastrointestinal disturbances affected 22% of the cohort (see Table 1 to compare prevalence by gender).

Regarding the findings on cardiac imaging studies, LVH affected 35% of the population and was observed more frequently in men (47% vs. 26%, *p* = 0.13), with mean maximum wall thicknesses of 16 vs. 10.5 mm by cardiac MRI (*p* = 0.11) and 13 vs. 9.5 mm by echocardiogram (*p* = 0.048) in men and women, respectively. Left atrial dilatation was similar in both genders (*p* = 0.33).

Cardiac magnetic resonance was performed only in 43% of all patients, and in a few of them, T1 mapping sequences were performed for deeper tissue characterization (Figure 3). Of those, 48% showed low native T1 mapping values, some of them even without evidence of myocardial hypertrophy; this finding was less common in men than in women (40% in men and 60% in women, *p* = 0.52) (Figure 3). On the other hand, late gadolinium enhancement (LGE) suggestive of focal fibrosis was observed in 31% of the studied patients; this finding was 1.5 times higher in men than in women (38 vs. 25%, *p* = 0.45) (Figure 3).

### 2.2. Biochemical Characteristics According to Gender

Seventy-two percent of carriers showed reduced or undetectable levels of α-GAL A enzymatic activity. This finding affected all men (37% undetectable and 61% reduced values), while only 50% of women presented reduced levels of enzymatic activity, and none showed undetectable activity.

Metabolite values such as Lyso Gb3 were found to be increased in 81% of cases; this finding tended to be more frequent in men than in women (94% vs. 71%, *p* = 0.086). Natriuretic peptides were slightly higher in men (292 vs. 129 pg/mL, *p* = 0.44).

### 2.3. Specific Treatment and Gender Differences

During follow-up, almost half of the cohort required some form of enzyme therapy. Twenty-seven percent of the women and 75% of the men received some type of replacement or accompanying treatment, with no differences between these strategies. The change in therapeutic strategy to chaperone was infrequent and was due to poor tolerance to the intravenous infusion of enzymes in all cases. The median age at initiation of enzyme therapy was 44.2 years (IQR 36–57), with a delay of almost half a year in initiating therapy since diagnosis in women compared to men.

### 2.4. Life Expectancy and Event-Free Survival

Nine percent of the cohort died during follow-up. The most frequent cause of death was related to end-stage renal failure or progressive heart failure (Table 1).

The combined incidence of serious cardiovascular and renal events such as stroke, need for dialysis, heart failure, or death in the entire cohort of carriers was 33% and a median event-free survival of 69 years (IQR 55–75 years). The Kaplan–Meier combined event-free survival is shown in Figure 4. There was a higher trend for these isolated events as well as for combined events in men (40% in men vs. 28% in women, *p* = 0.36) (Figure 3 and Table 2).

## 3. Discussion

The scarce evidence of the behavior of the p.Arg301Gln variant in the *GLA* [16] gene comes from case reports and is consistent with the findings of this study; we observed an intermediate evolution between classic and late-onset genetic variants, both in the extension of the affected organ as well as at the time of clinical manifestations. This is the largest international cohort of patients with this genetic variant in the *GLA* gene described to date.

Global penetrance was 63%, clinically affecting 80% of men and only 52% of women. Men presented symptoms on average 10 years earlier than women, with a shorter delay in diagnosis and initiation of specific treatment. In agreement with previous descriptions in nontypical variants of AFD, the mean time of symptom development was close to the fourth decade of life [17]. It is important to note that approximately 20% of the men were asymptomatic. This finding could be associated with the early and extensive family screening (see example in Figure 5), and the existence of a preclinical stage of the disease.

The diagnosis in males was more frequently associated with typical symptoms and suggestive abnormalities in complementary tests, while women were more frequently diagnosed by family screening. Since an active search strategy and family extended screening were applied, five patients (10%) were detected before the development of symptoms; therefore, these patients had to be excluded from the analysis of diagnostic delay. However, exploratory analysis showed no difference in the symptom’s onset in those with pre-symptomatic and post-symptomatic diagnosis.

Another distinctive finding to note is that, in contrast to what is mentioned in the literature, classical manifestations of the disease such as cornea verticillata, angiokeratomas, or acroparesthesias were observed in less than 20% of all patients (30% in men vs. 14% in women).

The prevalence of atrial fibrillation was higher than expected due to the individual age and comorbidities, and was almost four times higher than that described in other series, where it affects almost 4% of the general population [18].

Unlike other late-onset genetic variants on the *GLA* gene such as p.Phe113Leu (NM_000169.3 (*GLA*):c.337T > C), cardiac involvement was less severe and frequent in R301Q carriers [17]. However, the clinical manifestations began much earlier. Reports show that patients carrying the p.Phe113Leu variant usually develop symptoms at the age of 47, whereas in our cohort, men developed symptoms at 36. LVH affected 35% of the cohort and specifically 45% of men; this was much lower than that described in the literature, which reached almost 75% of the male population and almost 25% of women with nonclassical variants [18]. However, heart failure symptoms affected 29% of the general cohort and were more frequent in men than women (35% vs. 24%, *p*: 0.41), more than that reported in late-onset variants [19].

A low native T1 mapping value was observed in 8 of the 13 (61.5%) symptomatic patients and 2 of the 8 (25%) asymptomatic patients who underwent cardiac MRI at follow-up; in some of them, even in the absence of LVH. These values were far from those described in recent studies, showing this finding in 89% of patients with ventricular hypertrophy and 48% without ventricular hypertrophy [20].

Likewise, the prevalence of a late gadolinium enhancement in 31% of the population and almost half of the patients studied by cardiac MRI is in line with previous descriptions where this finding reached 25 to 50% of the carriers; in these reported cases, 20% were still in the pre-hypertrophic phase, and up to 75 to 100% involved the intramyocardial basal inferolateral segment according to the hypertrophy installed [17,19]. It is important to note that women more frequently presented with low T1 mapping while men more frequently presented with +LGE; this finding could be explained by the pseudonormalization of T1 mapping values due to the presence of fibrosis.

A lower percentage of patients with abnormal α-GAL A activity was observed with respect to those with Lyso Gb3 elevation. The discordance only affected females. This reflects the poor diagnostic utility of α-GAL A enzyme activity and the need to search for the genetic variant or metabolites in biological fluids as a confirmatory diagnostic method in women.

Eighty percent of the patients who suffered a significant clinical event had reduced enzyme activity levels, and a smaller proportion had undetectable levels. Although, to date, there is no direct relationship between the degree of α-GAL A activity and the severity of clinical involvement, in our study, no woman carrying the genetic variant and with normal enzyme activity suffered significant clinical events such as advanced renal failure, stroke, heart failure, or severe arrhythmias. In contrast, about 18% of all patients, some of them even with undetectable levels of enzyme activity, had no clinically relevant events at mid-term follow-up. This may reflect not only the incomplete penetrance of the disease, but also the short follow-up of this cohort and reinforces the concept of the limited place of biomarker-guided clinical and therapeutic follow-up in daily practice.

The stroke rate was 8%, similar that is described in international registries of patients with AFD [21,22], and much higher than that expected in the general population, even after adjusting groups by age [21,22].

When arrhythmic events were analyzed, the incidence of atrial fibrillation was similar to previous reports, although NSVT and SVT were more frequent than expected [16]. The same occurred with the need for device placement, pacemaker, or implantable cardiac defibrillator (ICD). This is striking, especially considering the young age of our patients. The high incidence of arrhythmias and the need for device implantation, which all together affected 20% of the population, could demonstrate the important role played by arrhythmias in the evolution of patients with AFD.

With respect to kidney disease, the proportion of patients with proteinuria was like that reported previously. Still, the need for dialysis was much higher than in other late-onset variants such as p.Phe113Leu, which usually shows an incidence of severe renal failure like the general population [13,19,23].

The cumulative incidence of death was 9% in the entire cohort, with males accounting for 15% and females for only 4%. Three deaths in men were associated with heart or end-stage renal failure, and the only death of a woman was not initially attributed to AFD but to valvular heart disease in the setting of severe aortic stenosis.

As previously described and as in other metabolic diseases, the genotype–phenotype correlation is not completely understood; many of these mutations are ‘private’, vary even within the same family, or can cause nonspecific complications common to other diseases. However, the detection of specific genetic variants such as p.Arg301Gln remains of clinical significance, not only because of the prognostic implications and the prediction of organ-specific damage detailed above, but also because only some of them such as the p.Arg301Gln variant are considered ‘amenable’ to treatment with oral chaperones. In our cohort, 48% of the patients required some type of specific treatment; more than half of them received oral migalastat every other day, while the rest received biweekly intravenous enzyme replacement therapy.

### Limitations

The enzyme activity was measured with different techniques in plasma or leukocytes. Likewise, due to the heterogeneity in the reference levels of α-GAL A activity and Lyso Gb3 concentration used by the different laboratories of the participating European centers and the poor performance of these markers for predicting the evolution of the disease in the follow-up of these patients, the absolute values of these markers were not detailed in this study and are only described categorically according to the literature as to whether they were in the normal, reduced, or undetectable range.

Although we have mentioned that the 20% rate of asymptomatic men may be due to preclinical diagnosis of a disease with nonclassical AFD, given the retrospective nature of this manuscript and the residual activity of α-GAL A levels, even men may not always present with relevant clinical manifestations and may be misdiagnosed as asymptomatic. In addition, since this study had a retrospective approach, the determination of symptoms or clinical events could be subject to recall or misclassification bias.

Only a small proportion of the cohort underwent cardiac MRI, so these findings may not be generalizable to all patients with the p.Arg301Gln *GLA* variant.

While all combined and isolated endpoints including death were seen more frequently in men than in women, our study was statistically underpowered for hypothesis testing, so the *p*-value was not interpretable. Nevertheless, these differences are clinically important.

As a final comment, it is important to note that although the main objective of the study was to describe the phenotype of this particular variant, given the X-linked inheritance, an exploratory analysis was performed comparing males and females. Although we found clinically relevant differences in biochemical, clinical, and imaging behavior, the absence of statistical significance in some variables may be due to the low statistical power as a result of the small number of patients.

## 4. Materials and Methods

This was an observational, international, and retrospective cohort case series study involving five centers dedicated to rare genetic diseases in Spain, Germany, and Switzerland. The study was approved by institutional ethics committees and conformed to the principles of the Declaration of Helsinki. Written informed consent was obtained from all participants.

We collected information related to the clinical expression and biochemical characteristics of a particular genotype of patients with AFD as well as all direct relatives who were screened for the same p.Arg301Gln genetic variant in the *GLA* gene, using NGS techniquewith a Novaseq 6000Dx platform, from Illumina^®^, San Diego, CA, USA. Regardless of the initial method used, the presence of the p.301Gln variant was confirmed by the sequencing technique in all positive cases. All patients had periodical follow-up, independently, whether they were symptomatic or not. The proportion of asymptomatic carriers and the penetrance were determined at the end of the study period.

We described the enzymatic activity of α-GAL A, urinary, or serum levels of LysoGb3, its metabolite, and the presence or absence of proteinuria. The α-GAL A activity was defined as undetectable at less than 5% and reduced when 25–30% of enzymatic activity was detected [8]. We analyze it in serum by fluorometric technique. Given the heterogeneity of the methods used to measure enzymatic activity, we described this variable categorically, considering the reference values of each center.

We recorded the characteristics of electrocardiograms and cardiac imaging such as echocardiograms (Vivid E90 echo from General Electric^®^, Boston, MA, USA) and cardiac MRIs (GeneralElectric^®^ Healthcare SIGNA™ Explorer 1.5 T). We analyzed the clinical penetrance, the time lapse from symptoms onset to diagnosis, and the development of relevant clinical events such as those affecting the central nervous system (stroke), heart (heart failure symptoms), and kidneys (need of dialysis).

Direct family members underwent routine molecular testing, regardless of clinical suspicion of AFD. Only two women were considered as obligate carriers, as several of their children were genetically positive for the *GLA* variant. In these cases, the clinical information was collected retrospectively from the electronic medical records from each center.

The severity of heart failure symptoms was defined following the New York Heart Association Class (NYHA) [24].

Given the hereditary nature of the disease and given that all clinically relevant events were recorded as part of the clinical history of each patient, these events were considered incidents. Follow-up time was considered from the first to the last visit with the inherited conditions team.

Statistical analysis. Continuous variables were expressed as mean ± standard deviation or median ± interquartile range, while categorical variables were expressed as absolute and relative frequencies.

Continuous data between groups were compared with the Student’s *t*-test if the distribution of the variables was normal or with the Mann–Whitney–Wilcoxon test if not. A comparison of categorical data was performed with the chi-square test. Survival was represented graphically using Kaplan–Meier curves. An exploratory analysis of gender differences in this genetic variant of the disease was performed. A value of *p* < 0.05 was defined as statistically significant, working with two-tailed tests. The STATA 13.1 program was used for the statistical analysis.

## 5. Conclusions

Patients carrying the p.Arg301Gln variant in the *GLA* gene have a high penetrance of cardiorenal involvement with the clinical onset of the disease in middle age. The combined incidence of cardiovascular and renal events affected 33% of our population and presented a decade earlier in men. We consider that the genotype–phenotype correlation is useful from a practical clinical point of view and for decision-making.

## Figures and Tables

**Figure 1 ijms-25-04299-f001:**
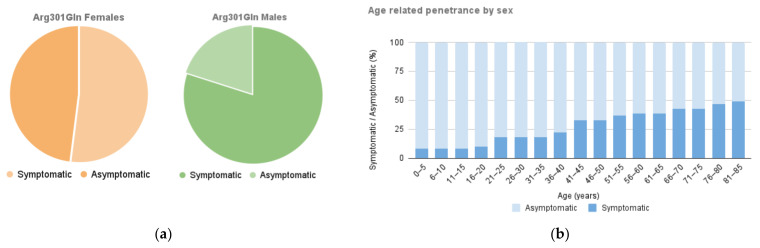
p.Arg301Gln carriers and clinically affected patients by gender (green: male, orange: female) (**a**) and age-related penetrance (**b**).

**Figure 2 ijms-25-04299-f002:**
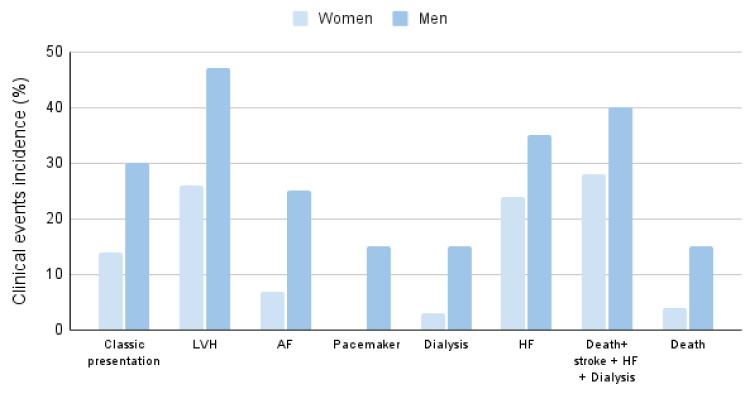
Clinical event incidence by gender (%).

**Figure 3 ijms-25-04299-f003:**
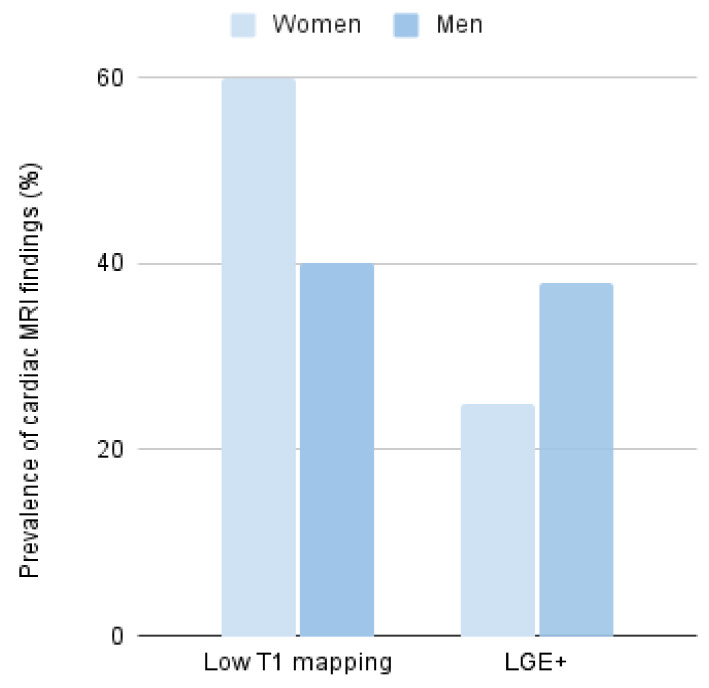
Cardiac MRI findings (T1 mapping and LGE) and differences by gender.

**Figure 4 ijms-25-04299-f004:**
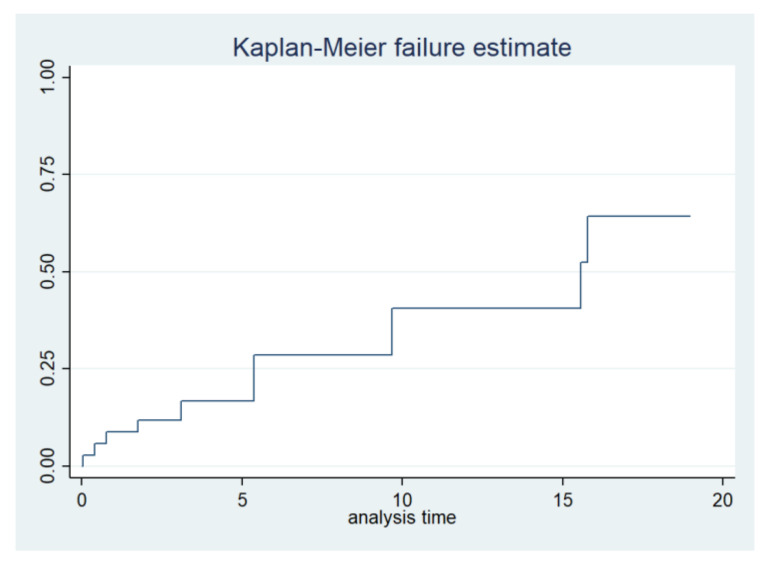
Combined Kaplan–Meier curve of events during follow-up.

**Figure 5 ijms-25-04299-f005:**
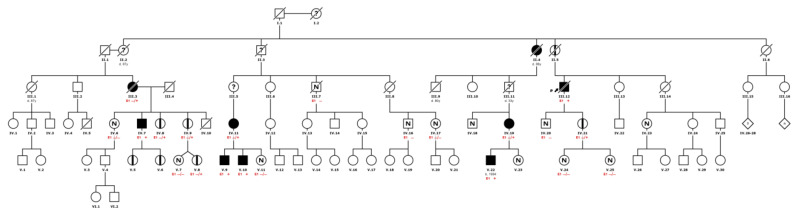
Example of family pedigree. AFD affected (filled square); *GLA* carrier—AFD not affected (partially filled square); unknown (?); not carrier (N).

**Table 1 ijms-25-04299-t001:** Clinical, biochemical, and cardiac imaging parameters characteristic of the entire cohort and differences by gender.

	General Cohort	Male	Female	*p*-Value
n (%)	49	20 (41%)	29 (59%)	
Age diagnostic, median (IQR)	40.91 (21.11, 55.90)	40.09 (19.35, 52.29)	41.56 (21.57, 57.19)	0.71
Age symptoms, median (IQR)	41.51 (21.01, 55.84)	37.29 (20.5, 49.14)	46.40 (21.52, 67.98)	0.32
Time diagnosis-treatment (years). Median (IQR)	0.91 (0.51, 1.74)	0.62 (0.40, 1.41)	1.45 (0.66, 4.29)	0.21
Symptomatic. n (%)	31 (63%)	16 (80%)	15 (52%)	0.044
Classical clinical manifestations. n (%)	10 (20%)	6 (30%)	4 (14%)	0.17
α-GAL A enzyme activity				
0: normal	11(28%)	0 (%)	11 (52%)	<0.001
1: reduced	21 (54%)	11 (16%)	10 (48%)	
2: undetectable	7 (18%)	7 (39%)	0 (0%)	
LysoGb3 increased. n (%)	30 (81%)	15 (94%)	15 (71%)	0.086
Neuropathic pain. n (%)	5 (10%)	3 (15%)	2 (7%)	0.36
Acroparesthesia. n (%)	11 (22%)	6 (30%)	5 (17%)	0.29
Hypohidrosis. n (%)	2 (4%)	0 (0%)	2 (7%)	0.23
CNS involvement. n (%)	6 (12%)	3 (15%)	3 (10%)	0.63
Cornea Verticillata. n (%)	2 (4%)	1 (5%)	1 (3%)	0.79
Hearing impairment. n (%)	7 (14%)	4 (20%)	3 (10%)	0.34
Angiokeratoma. n (%)	4 (8%)	2 (10%)	2 (7%)	0.7
Gastrointestinal manifestations. n (%)	11 (22%)	6 (30%)	5 (17%)	0.29
Proteinuria. n (%)	18 (38%)	9 (45%)	9 (32%)	0.36
Hypertension. n (%)	15 (36%)	9 (45%)	9 (32%)	0.36
Diabetes. n (%)	3 (6%)	2 (10%)	1 (3%)	0.35
PR < 121 ms. n (%)	9 (18%)	3 (15%)	6 (12%)	0.61
LVH EKG. n (%)	15 (36%)	8 (47%)	7 (28%)	0.21
LVH on echo/MRI. n (%)	16 (35%)	9 (47%)	7 (26%)	0.13
T-wave inversion	25 (58%)	14 (82%)	11 (42%)	0.009
Complete Bundle branch block. n (%)				
0: none	44 (90%)	15 (75%)	29 (100%)	
1: left	3 (6%)	3 (15%)	0 (0%)	0.008
2: right	2 (4%)	2 (10%)	0 (0%)	
AV Block. n (%)				
0: none	45 (92%)	18 (90%)	27 (93%)	
1º	4 (8%)	2 (10%)	2 (7%)	1
2º	0 (0%)	0 (0%)	0 (0%)	
3º	0 (0%)	0 (0%)	0 (0%)	
Atrial fibrillation of flutter. n (%)	7 (14%)	5 (35%)	2 (7%)	0.075
Pacemaker. n (%)	3 (6%)	3 (15%)	0 (0%)	0.031
NSVT. n (%)	4 (8%)	4 (20%)	0 (0%)	0.013
VT. n (%)	2 (4%)	2 (10%)	0 (0%)	0.087
Echo: max_wall_thickness (mm. Median (IQR)	11 (9, 17)	13 (9, 20)	9.5 (8, 13)	0.048
Echo LVEF %. Median (IQR)	63 (59, 70)	64.5 (58.5, 69.5)	62 (60, 71)	0.79
Diastolic pattern. n (%) 0: normal	14 (30%)	4 (21%)	10 (37%)	
1: impairment	21 (46%)	7 (37%)	14 (52%)	0.095
2: pseudonormal	10 (22%)	7 (37%)	3 (11%)	
3: restrictive	1 (2%)	1 (5%)	0 (0%)	
E/e. Median (IQR)	7.9 (6, 13)	0.5 (5.9, 14.32)	7 (6, 11)	0.38
Atrial dilatation (%)	22 (47%)	11 (55%)	11 (41%)	0.33
Global longitudinal strain %. Median (IQR)	19 (14.1, 20.62)	11.95 (8.55, 18.31)	20 (19, 21.4)	0.021
Cardiac MRI wall thickness (mm). Median (IQR)	11 (9, 17)	16 (10, 18)	10.5 (8, 15)	0.11
LGE +. n/MRI (%)	10/32 (31%)	6/16 (37.5%)	4/16 (25%)	0.45
Low T1 mapping. n/MRI (%)	10/21 (48%)	4/10 (40%)	6/11 (54%)	0.52
Treatment. n (%) 0: none	25 (52%)	4 (21%)	21 (72%)	0.003
1: agalsidase A	5 (10%)	4 (21%)	1 (3%)	
2: agalsidase B	6 (13%)	4 (21%)	2 (7%)	
3: migalastat	12 (25%)	7 (37%)	55 (17%)	
Death, Stroke, HF, or Renal Replacement. n (%)	16 (33%)	8 (40%)	8 (28%)	0.36
Death. n (%)	4 (9%)	3 (15%)	1 (4%)	0.17
Stroke. n (%)	4 (8%)	2 (25%)	1 (4%)	0.16
Heart Failure. n (%)	14 (29%)	7 (35%)	7 (24%)	0.41
Dialysis. n (%)	4 (8%)	3 (15%)	1 (3%)	0.15
NTproBNP (pg/mL). Median (IQR)	219 (70.5, 765)	292 (57, 899)	129 (74, 592)	0.44

**Table 2 ijms-25-04299-t002:** Organ compromise per patient and geographic area.

Patient	Gender	Country	Cornea Verticillate	Stroke	Proteinuria	Dialysis	HF or LVH	Pacemaker	NSVT/VT
1	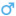								
2	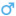								
3	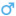								
4	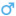								
5									
6									
7									
8									
9	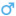								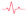
10	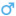								
11									
12	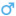			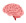					
13									
14									
15									
16				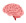					
17	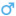								
18	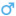								
19									
20									
21	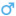							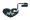	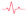
22									
23						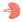			
24	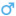					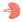			
25	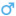					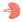			
26									
27									
28									
29									
30									
31	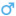								
32									
33	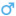		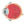	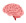				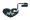	
34	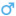								
35									
36									
37		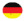							
38		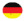							
39		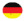							
40		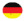							
41		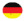							
42	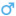	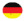							
43	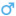	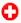		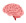					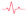
44		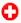							
45	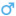	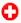							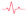
46		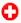	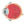						
47	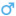	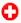				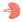		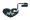	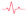
48		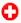							
49		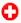							

## Data Availability

Data is contained within the article.

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
