# Peer review of "Phenotypic Expression and Outcomes in Patients with the p.Arg301Gln GLA Variant in Anderson–Fabry Disease"

_ijms, 2024, doi:10.3390/ijms25084299_

Round 1

Reviewer 1 Report

Comments and Suggestions for Authors

The paper is very well written and easy to read.

I have a few minor comments about the manuscript.

I suggest a brief explanation of the treatments available for the disease in the introduction, particularly using chaperones, and on which variants this is effective in, so when the text mentions that the Arg301Gln variant is amenable to chaperones, the reader is already familiar with it and understands how it compares to other variants.  

Please write “male” and “female” on top of the pizza plots in fig1 instead of only explaining in the legend.  This small change will make the figure more accessible to colorblind readers and to those who might print a grayscale copy.

Bar plots in the figures have no y axis annotation. In Fig1: Is the symptomatic/asymptomatic a ratio or percentage? Even though it looks obvious, it is best to clarify. Include the y axis annotation in the other plots as well.

Some figures have poor resolution in the file I am reviewing. Please make sure the figures are sharp enough to be read. Plus, the figure legend or title should not be part of the image (such as in fig 6), but should be part of the text to allow editing and sharpness of the letters. 

I recommend updating the nomenclature of F113L to the most up-to-date HGVS nomenclature throughout the paper. Also, Arg301Gln should be p.Arg301Gln according to HGVS. It is also important to include somewhere in the paper the reference sequence after which these variants were named, in case something changes in the future.

It would be very interesting if the authors could make a table summarizing their findings in comparison to the literature that was referenced in the discussion. This is not a must, but it would make the data easier to read and interpret.

Can you clarify the methods used for the genotyping and the biochemical analyses? Or cite appropriate references that describe the methods used? I understand these were performed in multiple centers, but I do think it is important to mention which technique was used in all of them, since this might vary and can bring different limitations to data interpretation. 

Finally, the sentence in Line 102 was cut short. Please review. 

Reviewer 2 Report

Comments and Suggestions for Authors

This observational, retrospective case series studies patients carrying the Arg301Gln variant in the GLA gene associated with Fabry disease with the aim to analyse the penetrance, clinical phenotype, and biochemical profile. This variant has been shown to be associated to the non-classical course of FD and is a migalastat-amenable variants. Therefore, natural history data seem of specific interest also with respect to therapy considerations.

The 49 carriers came mainly from Spain, in part also from Germany and Switzerland. Findings were a high penetrance, with predominantly cardiorenal involvement and clinical onset of the disease in middle age (men being more and earlier affected), thus, confirming the association of this variant to the non-classical form of FD.

I have mainly formal queries:

Two sentences are not correct (introduction):

-          ‘From the molecular point of view, the GLA gene consists of 7 exons and like other cardiomyopathies, ‘hot spots’ for point mutations have been identified in this gene, particularly those CpG and missense variants impacting the active site or key residues of the enzyme …’.  .. like in other genetic cardiomyopathies…?

-          ‘It ist is noteworthy that, like a few other genetic variants, this is considered ‘amenable’ to pharmacological treatment with oral chaperones’, ‘this’ meaning the Arg301Gln variant?!

References: Authors are missing in reference 12,  18 and 22. Citation of articles is not systematic in the references: sometimes only the first author followed by et al is given (obviously not the journal’s guideline), sometimes all authors
